# Assessment of P Wave Indices in Healthy Standardbred Horses

**DOI:** 10.3390/ani13061070

**Published:** 2023-03-16

**Authors:** Rebecca White, Laura Nath, Michelle Hebart, Samantha Franklin

**Affiliations:** School of Animal and Veterinary Sciences, The University of Adelaide, Roseworthy Campus, Roseworthy, SA 5371, Australia

**Keywords:** electrocardiography, ECG, cardiac, equine

## Abstract

**Simple Summary:**

Electrical activity of the heart is recorded using an electrocardiogram and represented as a series of waveforms. Measurements of the P wave component are used as non-invasive markers of remodelling and susceptibility to heart disease in humans. However, their usefulness has not been fully investigated in animals. The aim of this study was to measure P wave indices in healthy standardbred horses and investigate variables that might influence them. We found significant associations with exercise status as well as a positive correlation with exercise duration (number of years raced). These findings are similar to those reported in human athletes versus sedentary individuals. These changes are likely to be associated with the remodelling of the heart chambers that occurs with exercise training and may explain why these athletes are at increased risk of heart rhythm disturbances.

**Abstract:**

P wave indices are used as non-invasive electrocardiographic markers of atrial remodelling in humans. Few studies have investigated their use in animals. The aim of this study was to measure P wave duration and P wave dispersion (Pd) in healthy standardbred horses and investigate variables that might influence these measurements. A 12-lead electrocardiogram was recorded at rest and P wave indices were calculated in 53 horses. A general linear model was used to investigate the main effects: age, bodyweight, sex, resting heart rate, presence of a murmur, exercise status and the number of years raced. There were significant associations with exercise status for both the maximum P wave duration and Pd, with both values being increased in strenuously exercising versus non-active horses. Furthermore, a significant moderate positive correlation was identified between the duration of exercise (number of years raced) and both Pmax and Pd. No other significant associations were identified. These findings are similar to those reported in elite human athletes versus sedentary individuals. The increases in these P wave indices most likely occur due to prolongation and heterogeneity in atrial conduction time, which are associated with structural and electrical remodelling, and may explain the increased risk of atrial fibrillation in athletic horses.

## 1. Introduction

P wave duration and P wave dispersion (Pd) reflect the activation of atrial muscle and are used as non-invasive electrocardiographic markers of atrial remodelling in human medicine [1,2,3,4,5,6]. P wave dispersion is defined as the difference between the maximum (Pmax) and minimum (Pmin) P wave duration on a standard 12-lead surface electrocardiogram (ECG) [6]. Increased P wave duration reflects a prolongation in atrial conduction time, whereas increased Pd is a marker of heterogeneity in atrial conduction time [7]. These indices have been suggested to be sensitive predictors for the development of atrial arrhythmias, especially atrial fibrillation (AF) in human patients [3,4,5,6,7,8,9,10,11,12].

P wave duration and dispersion have also been shown to be increased in elite human athletes [13,14,15], and Pd is positively correlated with training duration [14,15,16]. These findings may explain the increased risk for the development of AF in this population. Numerous studies have identified an increased risk of AF in athletes compared with non-athletes, especially in those performing endurance exercise [17,18,19,20]. The mechanism for this increased risk is most likely related to the electrical and structural cardiac remodelling that occurs in response to athletic training [20,21,22,23,24].

Athletic horses, like human athletes, have been shown to experience cardiac remodelling in response to training [25,26,27,28]. Furthermore, similar to human athletes, AF is the most common arrhythmia affecting athletic performance in horses, especially racehorses, with a reported prevalence of 0.1–4.9% [29,30,31,32]. Atrial fibrillation may occur as a paroxysmal (PAF) or persistent event, with progression from PAF to persistent AF likely to occur over time in both humans [33,34] and horses [32,35]. The recurrence of AF is common both in horses with PAF [32] and horses with persistent AF that have undergone cardioversion [32,36]. This is a particular issue for racehorses that may be prevented from competing if they experience multiple episodes of post-race arrhythmia. A non-invasive measure to predict horses at an increased risk of developing AF would be extremely valuable.

Few studies have reported the use of P wave indices in veterinary medicine [37,38,39,40,41,42]. In dogs, Pd has been shown to be significantly increased in those with chronic valvular disease and disturbances of supraventricular conduction [38,42] and was also found to be a useful predictor of arrhythmia recurrence following electrical cardioversion of AF [39]. In horses, only one study has been performed to determine the values of Pd in healthy Silesian and Polish primitive horses [40].

The aim of this study was to measure P wave duration and dispersion in healthy standardbred (SB) horses and to investigate variables that might influence these measurements.

## 2. Materials and Methods

### 2.1. Study Population

This study was performed on 60 healthy SB horses with no history of cardiac arrhythmia, including 30 retired or unraced horses from the University of Adelaide’s teaching herd and 30 client-owned horses in race training.

### 2.2. Clinical Examination

Examinations were performed in the field (in small yards), both for horses from the teaching herd and client-owned racehorses. All horses underwent a brief clinical examination that included heart rate (HR), respiratory rate (RR), rectal temperature and thoracic auscultation. For teaching horses, bodyweight was recorded using electronic weigh scales, while those examined at racing yards had bodyweight estimated according to the following equation: girth^2^ × length [43]. Echocardiography was not performed.

### 2.3. ECG Recording and Analysis

Following the clinical examination, all horses underwent an ECG recording at rest using a wireless 12-lead Holter recorder (NR-1207-3; Norav Medical; www.noravmedical.com; accessed on 15 February 2023) attached to a surcingle. Horses were restrained by a handler, with a headcollar and lead rope, during the ECG collection to ensure that they stood quietly. Electrode placement was performed according to the modified method described by Hesselkilde et al. [44]:LA = on top of the left scapula;RA = on top of the right scapula;LL = left of midline, caudal to xiphoid process;RL = on top of right scapula next to RA;V1 = between superficial pectoral muscles;V2 = left ventral triceps;V3 = left thorax in 6th intercostal space, at level of the shoulder joint;V4 = left thorax in 6th intercostal space, at the level of the elbow joint;V5 = right thorax in 6th intercostal space, at the level of the elbow joint;V6 = right ventral triceps.

Continuous digital ECG recordings were obtained over a 5 min period. Subsequent analysis of ECG traces was performed using proprietary software (Norav Rest ECG; Norav Medical; www.noravmedical.com; accessed on 15 February 2023) using a paper speed of 100 mm/s and amplitude of 40 mm/mV. Manual measurement of P wave duration was performed across all 12 leads for five cardiac cycles at a time when the HR was lowest, without 2nd-degree atrioventricular block, and when there was minimal interference in the baseline. Each P wave was measured separately on each lead, using digital callipers, with a precision of ±2 ms. P wave duration was measured as the distance between the onset (positive or negative deflection from the isoelectric line) and the offset (return of the isoelectric line) [7,45]. The minimum (Pmin) and maximum (Pmax) P wave duration were identified for each complex from the 12 leads, and Pd was calculated as the difference between Pmax and Pmin. The mean Pmin, Pmax and Pd were then calculated for each horse.

### 2.4. Statistical Analysis

Statistical analysis was performed using SPSS Statistics Version 25, and graphs were created in Prism version 9. All data were tested for normality with histogram and predicted scatter plots, and data are presented as mean ± standard deviation (SD) unless otherwise specified. Pearson’s correlations were used to investigate associations between Pmin, Pmax and Pd. Variables that might be associated with a change in Pmin, Pmax and Pd were examined, for all horses, using a general linear model (GLM). The main effects considered in the analysis were age, bodyweight, sex, resting HR, presence of a murmur, exercise status and the number of years raced. Exercise status was categorized as “non-active” (retired or unraced horses), “low intensity” (horses in pre-training or early in their training preparation and not exercising maximally) and “high intensity” (horses that were race-fit/racing). All one-way and two-way interactions were tested. Only terms where *p* < 0.05 remained in the multivariable models and stepwise backward elimination was utilized. The sex distribution, age, bodyweight, and resting HR were compared between non-active (retired/unraced) horses and client-owned horses in race training using a chi-squared test (sex) or independent sample T-test (age, bodyweight, HR). The potential effect of prior race training on P wave indices was investigated in non-active horses by comparing values for Pmin, Pmax and Pd between those that had previously raced and those that had never raced, using independent sample T-tests. Finally, the effect of exercise duration (number of years raced) on P wave indices was further investigated by performing Pearson’s correlations using a subset of horses (n = 38) that excluded horses from the non-active group that had raced previously. Statistical significance was set at *p* < 0.05.

## 3. Results

### 3.1. Animals

The final analysis included 53 horses, with 7 horses excluded due to incomplete ECG data across all leads because of an artefact that made accurate measurement of P wave duration difficult in some leads. The included horses consisted of 32 females and 21 geldings, with a mean (SD) age of 8.5 (4.7) years (range: 2–18 years) and mean (SD) bodyweight of 476 (57) kg. Twenty-four horses were non-active (of which nine were unraced and fifteen had raced previously; the mean (SD) time out of racing was 6.8 (2.1) years), fourteen were classified as “low intensity” exercise and fifteen were classified as “high intensity” exercise. On clinical examination, the mean (SD) resting HR for all horses was 36 (4) bpm. Six horses (five in training and one non-active) had low-grade (≤grade 3/6) right-sided systolic murmurs, typical of tricuspid regurgitation, identified on cardiac auscultation. There were differences in the sex distribution, age, bodyweight, and resting HR for the non-active horses compared with horses that were in race training (Table 1).

### 3.2. P Wave Duration and Dispersion

Clear P waves were not identified in Lead I and were not included in the calculations. P wave indices were therefore calculated from the remaining 11 leads. The mean (SD) values of the entire population for Pmin, Pmax and Pd were 118.7 (15.5) ms, 162.3 (17.2) ms and 43.6 (9.4) ms, respectively.

There was a strong positive correlation between Pmax and Pmin (r = 0.842; *p* < 0.001). A similar but weaker correlation was found between Pmax and Pd (r = 0.449; *p* < 0.001). However, there was no correlation between Pmin and Pd (r = −0.104; *p* = 0.46).

#### 3.2.1. Variables Associated with P Wave Duration and Dispersion

The results of the univariable analysis from all 53 horses are presented in Table 2. There were no significant associations between Pmin and any of the measured variables. There was a significant association with exercise status for both Pmax and Pd, with both values being increased in strenuously exercising versus non-active horses (Figure 1a–c). In addition, the presence of a murmur and the number of years raced were associated with increased Pd. However, in the multivariable model for Pd, only exercise status remained significant.

#### 3.2.2. Effects of Prior Racing and Exercise Duration on P Wave Indices

There were no significant differences between non-active horses that had previously raced compared with those that had never raced for Pmin (115.6 (23.9) ms versus 114.8 (14.5) ms; *p* = 0.93), Pmax (155.2 (22.5) ms vs. 154.7 (18.7) ms; *p* = 0.95) or Pd (39.6 (7.1) ms versus 39.9 (7.0) ms; *p* = 0.93).

In a subset of 38 horses that excluded the 15 retired horses that had previously raced, significant positive correlations with exercise duration (number of years raced) were identified for both Pmax (r = 0.39; *p* = 0.02) and Pd (r = 0.41; *p* = 0.01) but not Pmin (Figure 2a,b).

## 4. Discussion

This study reports on P wave indices in healthy SB horses and investigates potential variables that might influence these measurements. The P wave durations recorded in this study were similar to those reported in SB horses in a previous study using the same 12-lead configuration [44]. In that study, mean (SD) values for P wave duration varied between 131 (14) ms and 158 (19) ms in each of the 12 leads, although Pmin and Pmax were not reported per se. P wave dispersion has not previously been described in SB horses but was reported in one study that examined Polish and Silesian horses [40]. The mean (SD) Pd reported in that study was slightly lower (30.2 (4.3) ms) compared with this study (43.6 (9.4) ms). These differences may be related to differences between breeds, exercise status or differences in the methodology used. In this study, P wave durations were measured over 11 leads, whilst in the study by Michlik et al., only 9 leads were used [40]. Previous studies in animals have used either six or nine leads to calculate the P wave indices [37,38,39]. However, it has been reported that the use of adjacent leads with shared vectorial orientation may provide greater sensitivity for distinguishing inhomogeneity of atrial activation [46], and it has been suggested that ECGs with ≤nine leads should be excluded from the analysis [9,45,46]. Nevertheless, no studies to date have correlated P wave indices measured in different leads or reported which leads are most useful to derive the calculated P wave indices. The positioning of the electrodes in this study was based on a previously published configuration, whereby the precordial electrodes were positioned ventral to the heart (except for V3) [44]. Placement of the electrodes in a more dorsal position (to be better optimized for the atria) may have increased the amplitude of the recorded P waves and led to improved accuracy in the measurement of P wave duration and dispersion.

The only variable that we found to have a significant association with P wave duration and dispersion, in this study, was exercise status. We found that non-active horses had significantly lower Pmax and Pd in comparison with horses that were racing. Another study, which recorded ECG parameters in a single lead (Lead II), has also identified a significant increase in P wave duration in trained versus untrained SB horses [47]. Contradictory results have been reported in the few studies investigating P wave indices in human athletes. Several have reported Pmax and Pd to be increased in elite human athletes compared with sedentary controls [13,14,15]. Furthermore, a positive correlation between training duration and Pd has been identified in humans, and these changes have been suggested to be associated with the cardiac remodelling that occurs in association with exercise training [14,15,16]. However, another study found no differences in P wave duration and dispersion between trained college athletes and control subjects [48]. Furthermore, Elliott et al. found no difference in P wave duration using a three-lead ECG, between recreational endurance athletes grouped according to lifetime training hours, although no control group was included [21]. We identified an association between the number of years a horse had spent racing and Pd using the univariable analysis. However, this did not remain significant in the final multivariable model. Over half of the non-active horses in our study had raced previously, and this may have influenced the results since the cardiac remodelling associated with exercise training may be reversible. Previously, it has been reported that deconditioning resulted in a reduction in cardiac dimensions and indices of cardiac function over a 12-week period in a group of 3–4-year-old SB horses [49]. We found no differences in P wave indices of non-active horses that had previously raced compared with those that had never raced, suggesting that reverse remodelling may have occurred in those horses that previously raced. When non-active horses that had previously raced were removed from the analysis, we found significant moderate positive correlations between exercise duration (in years) and both Pd and Pmax.

Exercise training induces structural, functional and electrical adaptations within the heart including atrial dilation, myocardial fibrosis and modulation of autonomic tone associated with heightened vagal activity [21,22,23,50]. These changes in the “athletic heart” are believed to contribute to the increased risk of arrhythmogenesis and development of AF [23,51,52]. Atrial fibrillation is the most common performance-limiting arrhythmia of horses [29,30,31,32] and human endurance athletes [18,19,53]. In human athletes, the risk of AF is associated with the cumulative volume of exercise training [50,54,55], and this has also been suggested in SB racehorses [31]. P wave duration quantifies the time required for atrial depolarisation and depends on the mass of atrial tissue to be activated; thus, when prolonged, it might be considered a useful indicator of atrial enlargement [1,41,56]. Atrial size is an important factor for the perpetuation of AF since it facilitates the development of re-entry circuits by increasing the conduction time [35,50,57]. However, whilst a prolongation in P wave duration is commonly associated with AF [4,5,12,46,58], only a few studies have related the increase in P wave duration to atrial enlargement per se [59,60]. In athletes, lifetime training hours were associated with the prolongation in P wave duration and an increase in left atrial volume in one study [16]. However, a recent multicentre study found that although left atrial enlargement was common in competitive human athletes, it was not associated with a significant modification in electrocardiographic indices including the P wave duration [61]. Furthermore, although an association was found between P wave duration and the left atrial to aortic ratio (LA/Ao) in dogs with cardiovascular disease, this was not considered to be a reliable indicator of left atrial enlargement [41]. Prolonged P wave duration that occurs independent of an increase in atrial dimensions may reflect abnormal intra- and inter-atrial conduction due to atrial remodelling. Recently, increased P wave duration has been shown to be independently associated with left atrial fibrosis and mechanical dysynchrony [62]. In addition to P wave duration, it is well accepted that changes in P wave morphology and dispersion also have the potential to provide information regarding the anatomical substrate predisposing to AF [1,4,63]. Increased Pd is a marker of heterogeneity in the atrial conduction time [7]. It is considered a sensitive marker for atrial remodelling and a predictor of PAF [2]. Increased Pd has been found to be an independent predictor of PAF in patients with normal left atrial size and is proposed to occur due to changes within the atrial microarchitecture that increase the heterogeneity in sinus impulse conduction, which may occur before changes in LA dimensions [64]. Associations between left atrial size and Pd have been identified in patients with normal sinus rhythm [65] and PAF [64]. However, other studies on athletes have found no such association with atrial size [13,14]. Echocardiographic measurements of atrial size and function were not performed in this study. Measurement and calculation of left atrial fractional area, left atrial fractional shortening and tissue doppler measurements are well described in horses and have been shown to be useful in the assessment of atrial remodelling and reverse remodelling in cases with induced and naturally occurring AF [66,67,68]. Further studies incorporating echocardiographic measurements of cardiac dimensions and atrial function should be performed to investigate the potential associations between atrial size, atrial structural remodelling, and P wave duration and dispersion in horses.

Exercise training is associated with changes in autonomic tone and a reduction in the resting heart rate in human athletes [16,21]. P wave duration is also influenced by the autonomic nervous system, with β-adrenergic stimulation shortening and β-blockade lengthening P wave duration, whilst parasympathetic blockade shortens P wave duration [69]. Metin et al. [13] identified a negative correlation between resting heart rate and Pd in elite female basketball players. However, Puerta et al. [14] found a positive relationship between resting HR and Pd in the elite athletes they studied. Resting HR in the horses in this study ranged from 32 to 40 bpm during the ECG data collection period and no associations were identified with P wave duration or dispersion. Recently, Nissen et al. reported that trained SB horses had lower resting heart rates and increased frequency of 2nd-degree AV block compared with untrained horses [47]. In contrast, we found a slightly lower HR in non-active horses compared with those in race training, although HR in both groups was higher than those reported by Nissen et al. In that study, ECG was monitored for 2 h whilst horses stood quietly in their stable. In our study, horses were handled for the brief 5 min ECG recording, which may have promoted sympathetic stimulation and hence, the measurements of HR may not reflect the true resting rate.

Studies in humans have also identified associations between P wave indices and age, sex and bodyweight. P wave duration increases with age [8,70]. Males have been found to have increased Pd compared with females [71], and there is also a positive correlation with bodyweight [13,71]. The associations with sex and bodyweight in human patients have been suggested to be related to atrial size since increased bodyweight is one of the most important determinants of atrial size [72] and males tend to be heavier than females. In dogs, increased Pd has also been associated with greater bodyweight but not sex differences [42]. We found no significant association between age, sex or bodyweight and P wave duration or dispersion in this study. Michlik et al. also found no association between Pd and age, sex or bodyweight in the horses that they studied [40]. Changes in cardiac size are expected to occur in foals and young growing horses. Both Young et al. and Buhl et al. reported increases in cardiac dimensions over time in young (2- and 3-year-old) racehorses undergoing athletic training [25,26]. However, no untrained age-matched controls were included, making it difficult to differentiate between training-related changes and those associated with growth and development. Further changes in cardiac size do not appear to be associated with age in mature animals [73,74]. Several studies have investigated potential associations between body size and echocardiographic dimensions in adult horses. Long et al. examined Thoroughbred racehorses with a wide range of body weights (342–648 kg) and found no correlation between bodyweight and any cardiac dimensions [75]. Similarly, Bakos et al. found little or no linear correlation between cardiac dimensions and bodyweight in a group of SB horses weighing between 350 and 490 kg [76]. Zucca et al. did find a weak but statistically significant association between bodyweight and some cardiac dimensions including left ventricular and aortic size but not left atrial size [73]. In contrast, Buhl et al. observed a strong positive correlation between left ventricular size and body weight in their study and also reported that male horses had significantly larger ventricles than female horses [26]. However, no measurements of atrial size were reported.

There were a number of limitations to this study. The 12-lead electrode configuration used may not have been optimized for an assessment of the atria. However, we wanted to use a method that had previously been validated in horses. P wave measurements were performed manually and by a single observer and hence, could be subject to error and/or bias. However, manual measurement of P wave indices using digital callipers is commonly accepted and reported in other studies [45,46]. Reliability of P wave indices measurements requires accurate determination of the P wave onset and offset. This may be challenging if there is interference in the baseline, for example, if the animal moves during data collection. It has been suggested that leads should be excluded from the analysis if the baseline noise exceeds 10 mV and/or the peak to isoelectric line P wave amplitude is less than 15 mV [45], and in this study, seven horses were excluded due to excessive baseline interference. There were significant differences in the age, bodyweight and sex distributions between the non-active and exercising horses included in our study, and these could potentially influence the results. The non-active horses were older and heavier, and there was a higher proportion of females. Whilst we found no associations between these parameters and the measured P wave indices, it is possible that there might be some differences due to changes in atrial size in association with these parameters, as discussed above. Nevertheless, despite the increased age and bodyweight of non-active horses, we found that they had lower Pmax and Pd in comparison with exercising horses, suggesting that exercise training was the most important factor influencing these P wave indices. The lack of echocardiographic measurements is a further limitation of this study and would be useful to incorporate in future studies to determine the effect of atrial size and remodelling on P wave indices. Finally, whilst horses used in this study were assumed to be healthy and had no known history of arrhythmia, it is possible that some horses may have experienced episodes of PAF that were not identified.

## 5. Conclusions

This study reports on values for Pmin, Pmax and Pd in apparently healthy SB horses. We identified significant associations between exercise status and duration of time in racing and Pmax and Pd. Further studies are required to investigate potential associations between these P wave indices and left atrial size and function and to determine whether they might be useful as non-invasive predictors of AF.

## Figures and Tables

**Figure 1 animals-13-01070-f001:**
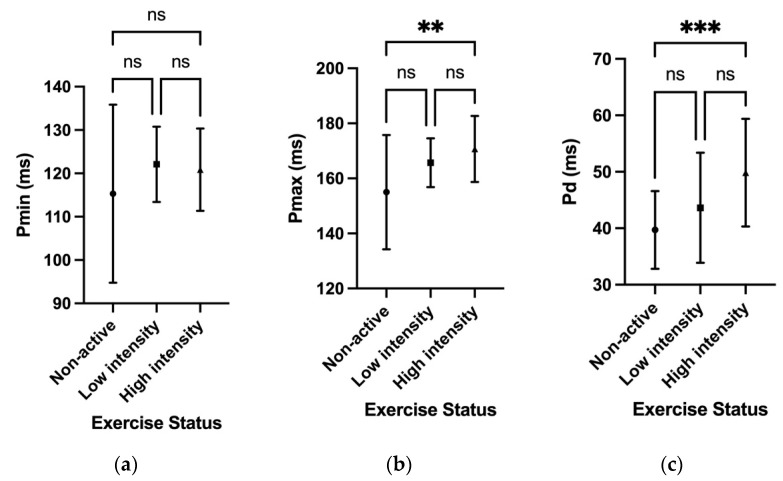
Mean ± SD values for: (**a**) Pmin, (**b**) Pmax and (**c**) Pd for horses according to exercise status (** *p* < 0.005); *** *p* < 0.001; ns not significant)**.**

**Figure 2 animals-13-01070-f002:**
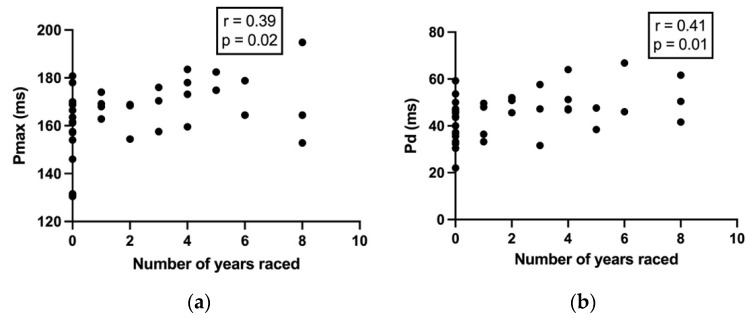
Relationship between exercise duration (number of years raced) and (**a**) Pmax and (**b**) Pd.

**Table 1 animals-13-01070-t001:** Baseline details for non-active horses compared with those in race training.

Parameter	Non-Active (n = 24)	Exercised (n = 29)	*p* Value
Sex:			
Male (gelding)	6 (25%)	15 (52%)	0.048
Female	18 (75%)	14 (48%)	
Age (years): Mean (SD)	12.4 (2.9)	5.4 (3.3)	<0.001
Bodyweight (kg): Mean (SD)	509 (41)	449 (54)	<0.001
Resting HR (bpm): Mean (SD)	35 (4)	37 (4)	0.03

**Table 2 animals-13-01070-t002:** Univariable linear regression of the relationship between each variable and Pmin, Pmax and Pd. Means (standard deviations) for fixed effects, Sex, Exercise Status and Murmur, regression coefficients **β** (standard error) for covariates (Age, Bodyweight, Resting HR and Years raced),

Variable (n)	Pmin		Pmax		Pd	
	Mean (SD)	*p* Value	Mean (SD)	*p* Value	Mean (SD)	*p* Value
Sex:						
Female (32)	120.0 (15.2)	0.463	161.6 (17.0)	0.717	41.6 (8.0)	0.056
Gelding (21)	117.7 (16.1)		163.4 (18.0)		46.0 (10.6)	
Exercise Status:						
Non-active (24)	115.3 (20.5)	0.355	155.0 (20.8)	0.012	39.7 (6.9)	0.003
Low intensity (14)	122.1 (8.7)		165.7 (8.9)		43.6 (9.8)	
High intensity (15)	120.9 (9.5)		170.7 (12.0)		49.8 (9.5)	
Murmur:						
No (43)	118.1 (16.1)	0.445	160.6 (17.5)	0.051	42.6 (9.0)	0.020
Yes (6)	123.3 (9.5)		175.1 (7.5)		51.9 (8.5)	

	**β (SE)**	***p* Value**	**β (SE)**	***p* Value**	**β (SE)**	***p* Value**
Age:	−0.424 (0.473)	0.358	−0.575 (0.507)	0.263	−0.151 (0.278)	0.591
Bodyweight:	0.023 (0.039)	0.547	0.019 (0.042)	0.650	−0.004 (0.468)	0.873
Resting HR:	0.437 (0.477)	0.175	0.986 (0.564)	0.092	0.264 (0.313)	0.402
Years raced:	−0.118 (0.835)	0.886	1.038 (0.902)	0.255	1.156 (0.468)	0.017

## Data Availability

Not applicable.

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
