# Peer review of "Assessment of P Wave Indices in Healthy Standardbred Horses"

_animals, 2023, doi:10.3390/ani13061070_

Round 1

Reviewer 1 Report

Dear Authors,

The original paper “Assessment of P wave indices in healthy Standardbred horses” could add some new information about the normal measurements and the factors able to influence P wave indices in healthy horses. The paper is well written, nevertheless, I have a few remarks and suggestions I would like the authors to consider.

1-Introduction:

L39-40: “These indices have been suggested to be sensitive… atrial fibrillation”. Please, precise this is in human medicine according to your references.

2-Materials and methods

L79: Please, could you describe the conditions of the ECG examination (in a stall after a short period of habituation for example) or on the field both for the teaching herd and the client-owned horses, since the variation in autonomous tone could influence your results.

L116-118: the selection of the population you used to test the influence of the parameter “years raced” is not clear. Could you please explain a bit more if you include horses from the control group who had never raced (only 9/24) in the materials and methods section and why you excluded those who had raced before, later in the discussion section.

3-Results:

L123: since age (with a wide range in your study: 2-18year-old) and sex have been shown to influence Pwave indices in previous studies and theses parameters are not correlated to P wave indices in your study, you could add the mean +/- SD for age and the sex distribution in control horses and in trained horses and maybe discuss this point in the discussion section, if relevant.

L144-145: Please could you precise for which parameters “exercise status “ was significant P duration, Pd or both?

L146: table 1: please add the number of horses includes in the statistical analysis for the parameter “years raced”  in the table because it is different from the rest of the study

L150-151: figure 1: the subhead of the figure is incomplete. Please add if the results were significant according to univariate or multivariate analysis

4-Discussion:

L194-195: “Over half of the non active horses had raced previously”, please could you discuss this point considering this could suggest the changes and remodeling of the atria (supposed to be the cause of P wave variations) with training could be reversible as it has been described for short duration atrial fibrillation (DeCloedt et al Vet J 2020).

L207-219: “P wave duration … a usefull indicator of atrial enlargement.” Morphological echocardiographic measurement like atrial size has been correlated to Body weight and even if you did not measure atrial size in your study, you did not find any correlation between Body weight and P wave indices as we could expect and this point may be discussed also.

L220-236:  P wave duration and dispersion and atrial remodelling: it could be interesting to propose further tests to assess functional remodeling (for example echocardiographic measurement and calculation of left atrial fractional shortening or left atrial fractional area change, or tissue doppler measurements well described in horses (Hessellilde et al BMC cardiovasc res 2019) to valid your hypothesis in a further study.

L262-273 you may develop more the limitations of your study including the high number of retired horses who have previously raced (if atrial remodelling could include some degree of fibrosis, is it fully reversible?) in your control group, the absence of objective evaluation of left atrial size, function or autonomic tone (Heart rate variability for example), and the inclusion of horses with a cardiac murmur on auscultation without echocardiographic assessment since this parameter was significant at least in the univariate analysis.

Reviewer 2 Report

I found the article about the assessment of P wave indices in healthy horses a very interesting work. It is novel, the methodology is sound, and overall the article is very well written with an interesting discussion. Therefore I support publishing. I only have little minor remarks.

Line 78: The P wave duration and morphology is heavily heart rate depended in horses, did you only record or measure during a specific heart rate range? If so please mention. Please also mention the range in the results, it is now only mentioned in the discussion as new information (or I have overlooked it in the results section).

Line 99: Please clarify this sentence that you measure each P wave complex separately on each lead. When I first read the sentence I had the impression you measured over all 12 leads at once. In addition, is 40mm/mV not extremely high?

Line 151: Something went wrong here.

Line 171: In the current lead configuration the precordial electrodes are placed ventrally of the heart (except V3) , do you think that this has impacted the sensitivity of the test? Wouldn’t placing the electrodes at the level of the heart increase the sensitivity?

Reviewer 3 Report

The manuscript is describing the assessment of p-wave indices in healthy trained and untrained standardbred racehorses in order to detect possible effects of training. The idea behind it is not new, but has not been investigated has here described before. The limitations of the study, namely no echocardiography and former trained horses in the untrained group has been addressed by the authors in the limitations. 

I only have minor comments:

Formal/writing: page 1, line 39 (introduction): 

 These indices have been suggested to be sensitive predictors for the development of atrial arrhythmias, especially atrial fibrillation

MM: study population:

-       Sedentary horses: retired race horses. How long were the horses out of training?

Discussion: at times unclear if the cited studies are from veterinary medicine (dogs) or humans or lab animals. Please clarify. 
